# Genome-Wide Identification of Wheat Gene Resources Conferring Resistance to Stripe Rust

**DOI:** 10.3390/plants14121883

**Published:** 2025-06-19

**Authors:** Qiaoyun Ma, Dong Yan, Binshuang Pang, Jianfang Bai, Weibing Yang, Jiangang Gao, Xianchao Chen, Qiling Hou, Honghong Zhang, Li Tian, Yahui Li, Jizeng Jia, Lei Zhang, Zhaobo Chen, Lifeng Gao, Xiangzheng Liao

**Affiliations:** 1Institute of Hybrid Wheat, Beijing Academy of Agricultural and Forestry Sciences (BAAFS), Beijing 100097, China; 2Institute of Crop Sciences, Chinese Academy of Agricultural Sciences, Beijing 100081, China; 18500830907@163.com (D.Y.); gaolifeng@caas.cn (L.G.); 3Chengdu Institute of Biology, Chinese Academy of Sciences, Chengdu 610041, China

**Keywords:** GWAS, haplotype, QTL, SNP array, stripe rust, wheat

## Abstract

Stripe rust, caused by *Puccinia striiformis* f. sp. *tritici* (*Pst*), threatens global wheat production. Breeding resistant varieties is a key to disease control. In this study, 198 modern wheat varieties were phenotyped with the prevalent *Pst* races CYR33 and CYR34 at the seedling stage and with mixed *Pst* races at the adult-plant stage. Seven stable resistance varieties with infection type (IT) ≤ 2 and disease severity (DS) ≤ 20% were found, including five Chinese accessions (Zhengpinmai8, Zhengmai1860, Zhoumai36, Lantian36, and Chuanmai32), one USA accession (GA081628-13E16), and one Pakistani accession (Pa12). The genotyping applied a 55K wheat single-nucleotide polymorphism (SNP) array. A genome-wide association study (GWAS) identified 14 QTL using a significance threshold of *p* ≤ 0.001, which distributed on chromosomes 1B (4), 1D (2), 2B (4), 6B, 6D, 7B, and 7D (4 for CYR33, 7 for CYR34, 3 for mixed *Pst* races), explaining 6.04% to 18.32% of the phenotypic variance. Nine of these QTL were potentially novel, as they did not overlap with the previously reported *Yr* or QTL loci within a ±5.0 Mb interval (consistent with genome-wide LD decay). The haplotypes and resistance effects were evaluated to identify the favorable haplotype for each QTL. Candidate genes within the QTL regions were inferred based on their transcription levels following the stripe rust inoculation. These resistant varieties, QTL haplotypes, and favorable alleles will aid in wheat breeding for stripe rust resistance.

## 1. Introduction

Bread wheat (*Triticum aestivum* L.) is one of the most crucial food crops globally and the second most widely grown food crop in China. Ensuring an adequate grain supply for the growing population is of utmost importance for food security. However, abiotic and biotic stresses present major challenges to wheat production. Stripe rust (*Yr*), caused by the fungus *Puccinia striiformis* f. sp. *tritici* Erikss. (*Pst*), seriously threatens global wheat production. It generally leads to 5–25% yield losses, and, in some cases, the losses can reach 70% [1]. In China, devastating stripe rust epidemics occurred in 1950, 1964, 1990, and 2002 [2]. In 2017, the disease affected 1.65 million hectares of wheat planting area across 12 provinces [3].

Wheat resistance to stripe rust can be classified as all-stage resistance (ASR) or adult-plant resistance (APR). ASR is detectable at the seedling stage and remains effective throughout the wheat development. It is usually controlled by a single major gene, which is often race-specific and qualitatively inherited. However, due to its specific nature and the strong selection pressure exerted by evolving pathogen populations, ASR is often overcome [4]. Consequently, ASR usually becomes ineffective within 3–5 years. In contrast, APR is typically controlled by multiple genes, each with a minor or partial effect, and it is highly expressed at the adult-plant stage. These genes do not confer immunity or high resistance and are known as horizontal, partial, race-nonspecific, slow-rusting, or durable resistances [5,6].

To date, more than 80 *Yr* genes for stripe rust resistance have been formally designated [7], and more than 100 temporarily named *Yr* genes and numerous QTL have been reported in wheat [8,9]. Most of these genes and QTL are valuable for breeding programs, and many, especially those identified in the past decade, have been used in marker-assisted selection. Nevertheless, most of the identified genes are race-specific. They interact with the pathogen following the gene-for-gene model and trigger hypersensitive reactions, rendering them ineffective against newly prevalent *Pst* races [1,10]. Pyramiding effective and race-nonspecific resistance genes into wheat varieties is an essential strategy to confer resistance against emerging *Pst* races [11,12].

China is regarded as a unique epidemiological stripe rust zone and the world’s largest independent epidemic region in terms of the wheat area affected by the disease [8,13]. By 2016, 34 *Pst* races, designated CYR1 to CYR34, had been identified in China using diverse sets of stripe rust differentials. Newly evolved *Pst* races (e.g., CYR34) exhibit broader virulence spectra, overcoming most known resistance genes [14,15,16]. Only genes *Yr5*, *Yr15*, and *Yr61*, along with several tentatively named genes, are effective against CYR34 [17]. Thus, new genes that are resistant to CYR34 are needed for wheat breeding.

The genome-wide association study (GWAS), an effective method of investigating the genetic variation underlying complex traits through a correlation analysis, has been successfully applied to explore the genes related to stripe rust resistance in wheat [18,19,20,21,22,23,24]. In this study, 198 wheat cultivars were evaluated for resistance to *Pst* at the seedling stage in a greenhouse and at the adult-plant stage in the field. We conducted a GWAS using the wheat 55K SNP array. The objectives of this study were to identify new resistance loci to *Pst* races and tightly linked SNPs that can be used to improve wheat stripe rust resistance in future breeding programs.

## 2. Results

### 2.1. Stripe Rust Response at the Seedling Stage

The IT and DS of 198 cultivars to CYR33 and CYR34 at the seedling stage were recorded in two replications (Appendix A). The phenotypic performance ranged from 0 to the maximum of 4 in infection types (ITs) and from 1 to 80% in disease severity (DS) (Figure 1a). A significant correlation was detected between the pairwise IT and DS values for both CYR34 and CYR33 infections (Figure 1b). The correlation coefficient for CYR33 (*r* = 0.37~0.60, *p* < 0.001) was lower than that for CYR34 (r = 0.72~0.91, *p* < 0.001), indicating different responses of accessions to CYR34 and CYR33.

Of the 198 accessions, 29 (14.6%) and 85 (42.9%) were resistant to CYR33 and CYR34 (DS ≤ 20%, IT 0–2), respectively (Appendix A). Nineteen accessions, including 17 Chinese modern cultivars, one from the USA (GA081628-13E16), and one from Pakistan (Pa 12), exhibited resistance to both *Pst* races. Among the 17 resistant cultivars widely grown in China, Yumai13, Zhengpinmai8, Fengdecun1, and Xiaoyan54 showed immune or nearly immune responses to both CYR33 and CYR34 (IT 0–0, DS < 1%) based on the classification standard of GB/T 1443.1-2007 [25].

### 2.2. Stripe Rust Response at the Adult-Plant Stage

The stripe rust responses of the 198 accessions were also evaluated in the field in three environments (CD18, CD19, and CD20) using a mixture of seven *Pst* races prevalent in China. The phenotypic performance ranged from 0 to the maximum of 4 in IT, and from 0 to 100% in DS (Figure 1a). Significant correlations (*r* = 0.67~0.94, *p* < 0.001) were observed for the IT and DS values among the three environments at the adult-plant stage (Figure 1b). In total, 41 accessions exhibited resistance to the mixed *Pst* races (Appendix A). Three accessions from the USA (GA081628-13E16, LA05032D-10, and LA06067C-P20) showed nearly immune responses to stripe rust with DS ≤ 1% and IT 0. Fifteen accessions had high resistance (DS ≤ 10% and IT ≤ 1), with three from the USA, three from Pakistan, and nine from China. If the DS range was set from 10% to 20%, and the IT range was set from 1 to 2 as moderately resistant (MR), an additional 23 accessions were identified, including six from Pakistan. Two accessions (St2422/464 and St1472/506) from Italy that are widely used in breeding in China showed moderate resistance to the mixed *Pst* races.

The panel had a higher frequency of resistance at the adult-plant stage (41 accessions) than at the seedling stage (19 accessions). Throughout the entire development, five wheat accessions from China (Chuanmai32, Lantian36, Zhoumai36, Zhengpinmai8, Zhengmai1860), one USA accession (GA081628-13E16), and one Pakistani accession (Pa12) demonstrated stable resistance to *Pst* races prevalent in China.

### 2.3. Population Structure, Genetic Diversity, and LD

After filtering, 44,759 SNP markers were used for the genetic analysis. The 198 accessions were grouped into two major clusters (G1 and G2) based on the SNP data, with G1 further divided into two subgroups and G2 further divided into eleven subgroups (Figure 2a). Most of the accessions in G1 were derivatives of Zhoumai9 and Aikang58 (Appendix A). In G1, subgroups G1-1 and G1-2 each contained 14 accessions. In the tests against the mixed races, CYR34, and CYR33, the number of resistant cultivars detected in G1-1 was 5, 11, and 5, respectively, while in G1-2, it was 3, 8, and 2. Therefore, at both the adult-plant and seedling stages, the frequency of resistant accessions in G1-1 was slightly higher than that in G1-2. Subgroup G2-2-1 mainly consisted of cultivars from Pakistan, which showed resistance to stripe rust at the adult-plant stage. Five accessions clustered in G2-6 showed high resistance to CYR34 at the seedling stage but were susceptible to stripe rust at the adult-plant stage, and three of them are pedigrees from Lovrin10. All seven American accessions were clustered in G2-7, and six of them showed high-resistance responses (IT 0~1) at the adult-plant stage (Figure 2a, Appendix A).

Based on the squared correlation coefficients (*r*^2^s) of all marker pairs among the 44,759 SNP markers, the extent of the LD decay in the 198 accessions was estimated, and the average LD decay rate was plotted against the physical distance. At the genome-wide level, when *r*^2^ ≥ 0.30, the LD decay distance was greater than 5.3 Mb, which was subsequently used as the confidence interval for significantly associated loci (Figure 2b).

### 2.4. GWAS and Haplotype Analysis for Stripe Rust Resistance

A GWAS using a mixed linear model identified 101 SNP markers that were significantly associated with stripe rust resistance. Specifically, 24 SNP markers were associated with resistance to CYR33 at the seedling stage, 50 were associated with resistance to CYR34 at the seedling stage, and 27 were associated with resistance to the mixed *Pst* races at the adult-plant stage. Considering the LD decay distance (±5.3 Mb), the MTA SNPs were clustered into 14 QTL, of which 11 were detected at the seedling stage and 3 at the adult-plant stage (Table 1). These 14 QTL mapped to chromosomes 1B, 1D, 2B, 6B, 6D, 7B, and 7D, explaining phenotypic variances (*R*^2^s) ranging from 6.04% to 18.32%. When compared with the previously reported *Yr* genes and QTL based on the CS reference genome, nine QTL, including *Yr.baafs-1B.1*, *Yr.baafs-1B.2*, *Yr.baafs-1B.3*, *Yr.baafs-1B.4*, *Yr.baafs-1D.1*, *Yr.baafs-1D.2*, *Yr.baafs-2B.1*, *Yr.baafs-2B.2*, and *Yr.baafs-7B*, were potentially novel loci. The remaining five QTL might be identical to the previously reported stripe rust resistance QTL or genes.

#### 2.4.1. QTL Resistant to CYR33 at the Seedling Stage

Four QTL significantly associated with resistance to CYR33 were detected based on the DS and/or IT records. These QTL were located on chromosome 2B (3) and chromosome 7B (1), explaining 6.1–17.2% of the phenotypic variance (Table 1).

*QYr.baafs-2B.1* was located within an 8 Mb interval (643.69–655.52 Mb, Figure 3a). A previously reported adult-plant resistance QTL, *QYrSM155.1*, was also located in this interval [26]. In *QYr.baafs-2B.1*, seven SNP markers significantly associated with CYR33 resistance were identified, forming 12 major haplotypes (Appendix A). Approximately 85.9% of the accessions carried haplotypes *QYr.baafs-2B.1-Hap1* to *Hap5*, with mean DS and IT values of 32.9% and 2.8, respectively. These results indicated that most of the Chinese commercial wheat varieties used in this study were moderately susceptible to the CYR33 race.

Among the identified haplotypes, *QYr.baafs-2B.1-Hap6* conferred the strongest resistance to CYR33 (Figure 3b). Six accessions carrying this haplotype exhibited strong resistance, including four immune (Zhengpinmai8, Fengdecunmai1, Shi4185, and Yumai13) and two moderately resistant cultivars (Aikang58 and Guomai9) to CYR33. Aikang58, Zhengpinmai8, Fengdecunmai1, and Guomai9 are derivatives of Zhou8425B, which was previously reported to have high resistance to stripe rust [31]. Zhou8425B carries multiple stripe rust resistance genes, including *YrZH84*, *YrZH84.2*, *YrZH22*, and *Yr30*. Additionally, Shi4185 and Yumai13 are derivatives of Lovrin10 and Funo, respectively. These parental varieties have been widely used in Chinese wheat breeding programs and are well-known for their resistance to stripe rust [32].

Among the seven SNP markers associated with a CYR33 response in *QYr.baafs-2B.1,* three peak SNPs (AX-110393982, AX-108773165, and AX-109974803) were identified within a 683 Kb region. These SNPs formed a haplotype block (Figure 3c), which encompasses eight high-confidence genes (*TraesCS2B02G452200*~*452900*). Five genes of these (*TraesCS2B02G452200* to *TraesCS2B02G452500*, and *TraesCS2B02G452900*) were expressed in the root, leaf, young spike, and developing grain tissues in Chinese Spring (Appendix A, data derived from WheatOmics) [33].

Upon further annotation, it was discovered that *TraesCS2B02G452300* contains a plastid lipid-associated protein PAP/fibrillin domain, while *TraesCS2B02G452400* encodes a receptor protein kinase with a leucine-rich repeat N-terminal domain. These protein families have been previously linked to disease resistance, especially the leucine-rich repeat kinase family. For instance, *TaFBN4* (*TraesCS4A02G272000*), encoding a plastid lipid-associated protein PAP/fibrillin domain, has been reported to be essential for wheat resistance against *Pst* race CYR33 [34]. The gene *TraesCS2B02G452300* was named *TaPAP8* and showed a high sequence similarity to its homologs in wheat (>90%) and other monocots (>80%). However, its sequence differed distinctly from *TraesCS4A02G272000* (Appendix A). Moreover, transcriptional data from WheatOmics [33,35] indicated that the expression patterns of *TaFBN4* and *TaPAP8* were similar in their responses to infection by *Pst* race CYR31 (Appendix A). These results suggest that *TraesCS2B02G452300* is a promising candidate gene underlying *QYr.baafs-2B.1*.

#### 2.4.2. QTL for Resistance to CYR34 at the Seedling Stage

Seven QTL conferring resistance to CYR34 were identified in both the IT and DS trials. These QTL were distributed across chromosomes 1BS (3), 1DS (2), 6DL(1), and 7DS(1). Six of these loci were regarded as major QTL, each accounting for over 10% of the phenotypic variance (Table 1).

##### QTL on Chromosome 1B

Three QTL located on chromosome 1B conferred resistance to CYR34. *QYr.baafs-1B.1*, *QYr.baafs-1B.2,* and *QYr.baafs-1B.3* were positioned on the short arm, within the intervals of 11.4 Mb, 6.3 Mb, and 6.7 Mb, respectively (Table 1). These QTL explained 8.23–18.32% of the phenotypic variance. Based on CS v1.0, *Yr15* (*Wtk1*) was mapped to 6BS, and its homolog *TraesCS1B01G079900* was located between 62,409,160 bp and 62,419,621 bp, 5 Mb proximal to *QYr.baafs-1B.1*. For *QYr.baafs-1B.1*, *-1B.2*, and *-1B.3*, seven, eight, and seven haplotypes were identified, respectively. The accessions carrying *QYr.baafs-1B.1-Hap2*, *QYr.baafs-1B.2-Hap2*, and *QYr.baafs-1B.3-Hap1* showed high resistance to CYR34. These resistance haplotypes were present in 38 accessions, all demonstrating resistance to CYR34 (IT 0 or 0 and DS 1%, Appendix A).

The *QYr.baafs-1B.2* locus, which explained 18.32% of the phenotypic variance, was located within the 90.6–96.9 Mb interval. According to CS v1.0, the interval contains 50 high-confidence genes and 6 low-confidence genes. Based on the transcriptome data [33,35], we found that three genes (*TraesCS1B02G090500*, *TraesCS1B02G091500*, and *TraesCS1B02G091900*) had their expression levels increased by more than double following the inoculation with *Pst* race CYR31 (Appendix A). *TraesCS1B02G090500* encodes a glutamate–cysteine ligase (GSH1) associated with a defense response against bacterial and fungal infections [36]. *TraesCS1B02G091500* encodes a bZIP transcription factor, which acts as a positive regulator in resistance to CYR32 [37]. *TraesCS1B02G091900* encodes cysteine-rich receptor-like protein kinase 2 (TaCRK2), involved in wheat rust resistance [38,39]. These genes are potential candidates for *QYr.baafs-1B.2*.

##### QTL on Chromosome 1D

Two QTL conferring resistance to CYR34 were mapped to chromosome 1D in both the IT and DS trials (Appendix A). The haplotypes *Yr.baafs-1D.1-Hap3* and *Yr.baafs-1D.2-Hap2* demonstrated significantly greater resistance to CYR34 infection compared to the other haplotypes. Specifically, these haplotypes exhibited IT values below 1 and DS values less than 10% (Appendix A). Among the four SNP markers strongly associated with CYR34 resistance in the *QYr.baafs-1D.2* region, AX-94555465 is located within the gene *TraesCS1D02G112300.1.* This gene encodes 6-phosphofructo-2-kinase/fructose-2,6-bisphosphatase, a crucial regulatory enzyme involved in the primary carbohydrate metabolism of photosynthetic plant tissues. Based on the transcriptome data [33,35], the expression level of *TraesCS1D02G112300.1* was found to be upregulated after the inoculation with stripe rust and powdery mildew (Appendix A).

##### QTL on Chromosome 6DL

*QYr.baafs-6D* was mapped to a 7.8 Mb interval on chromosome 6DL (465.05–472.82 Mb) (Table 1). Two SNP markers (AX-109362174 and AX-110999889), positioned at 467.5 Mb, explained up to 15% of the phenotypic variance at both the DS and IT trials. This locus overlapped with *QYrCL.sicau-6DL*, a previously identified adult-plant resistance QTL [29]. Three haplotypes formed for *QYr.baafs-6D,* and *QYr.baafs-6D-Hap3* was present in 32 accessions showing immunity or near immunity to CYR34 (Appendix A).

##### QTL on Chromosome 7DS

*QYr.baafs-7D* was mapped to a 100.7–104.7 Mb interval on the short arm of chromosome 7D. This locus explained 7.49–9.03% of the phenotypic variance for stripe rust resistance (Table 1). Notably, *QYr.baafs-7D* co-located with a previously reported adult-plant resistance QTL, *QYr.rcrrc-7D* [30]. Six SNP markers, namely AX-108860147, AX-109369183, AX-110392085, AX-111517979, AX-111855949, and AX-111687163, were significantly associated with stripe rust resistance. These markers defined three major haplotypes (Appendix A). Sixty-three accessions with *QYr.baafs-7D.1-Hap1* displayed immunity or near immunity to CYR34 (DS < 1% and IT 0 or 0). However, across all of the trials, the differences in the resistance effects among the haplotypes were not statistically significant. According to the WheatOmics data [33,35], within the *QYr.baafs-7D* region, the gene *TraesCS7D02G155100*, which encodes a protein kinase, showed upregulated expression levels at 24 h post inoculation (hpi) when challenged with the CYR31 and CYR32 races (Appendix A).

#### 2.4.3. QTL Resistant to Stripe Rust at the Adult-Plant Stage

Three QTL associated with resistance to the mixed *Pst* races at the adult-plant stage were identified on chromosomes 1BL, 2BL, and 6BS, with one locus located on each chromosome. These three APR QTL had relatively minor effects on stripe rust resistance, accounting for 6.04–9.27% of the phenotypic variations. Notably, *QYr.baafs-2B.2* and *QYr.baafs-6B* overlapped previously reported loci (Table 1).

Based on the CS v1.0 reference genome, 29 high-confidence genes are annotated in the *QYr.baafs-1B.4* region (487.94–491.08 Mb). Among these genes, *TraesCS1B01G281200.1* exhibited a positive response to both the stripe rust and powdery mildew infections (Appendix A). *TraesCS1B01G281200.1* encodes a transmembrane protein, and a previous study [40] has reported that such proteins are involved in plant disease resistance. Four SNP markers in the *QYr.baafs-1B.4* region (AX-110930185, AX-109885265, AX-108802815, and AX-108900982) were significantly associated with stripe rust resistance at the adult-plant stage. These four SNPs formed four haplotypes. *QYr.baafs-1B.4-Hap4* was present in five accessions that showed the highest susceptibility to mixed *Pst* infections at the adult-plant stage (Appendix A).

*QYr.baafs-2B.4* was positioned within a 7.2 Mb interval (ranging from 780.02 to 787.23 Mb) and overlapped with *qNV.Yr-2B.3*, a locus previously reported in Vavilov wheat [6]. *QYr.baafs-6B* was mapped between the SNP markers AX-110198445 (73.54 Mb) and AX-110360466 (91.01 Mb), overlapping with two APR QTL, *QYr.sicau-6BS* [27] and *QYr.nwafu-6BS.3* [28]. For *QYr.sicau-6BS,* 14 SNP markers associated with resistance formed nine haplotypes. Among these, *QYr.sicau-6BS-Hap5* exhibited the highest resistance to the mixed *Pst* infections (Appendix A). Four of the eight accessions carrying *QYr.sicau-6BS-Hap5* showed remarkable resistance phenotypes: LA06067C-P20 exhibited near immunity, with an IT of 0 and a DS of less than 1%, while Chuannong 19, Chuanmai 55, and Zhengmai 1860 showed resistance, with IT values below 1 and DS values under 20%. On chromosome 6BS, APR gene *Yr78* was finely mapped between CDM88 (101.735 Mb) and CDM103 (112.898 Mb) [41]. This mapping position is approximately 10 Mb proximal to *QYr.sicau-6BS*, suggesting that *QYr.sicau-6BS* likely represents a different APR gene.

#### 2.4.4. Distributions and Frequency of Favorable Haplotypes for Stripe Rust Resistance

The frequency of accessions carrying FHs was evaluated to provide valuable guidance for breeding programs. Among the 14 QTL identified, three QTL detected at the adult-plant stage (*Yr.baafs-1B.4*, *Yr.baafs-2B.4*, and *Yr.baafs-6B*) exhibited weak associations with dominant resistance haplotypes. This phenomenon can likely be attributed to their relatively small contributions to the phenotypic variance (Appendix A). Only four accessions, LA06067C-P20, Chuannong 19, Chuanmai 55, and Zhengmai 1860, were found to carry a FH of *Yr.baafs-6B*. Among them, Chuannong 19 and Chuanmai 55 demonstrated high resistance to the mixed *Pst* races that are prevalent in China. Across three different field environments at the adult-plant stage, these two varieties exhibited a DS of less than 10% and an IT of less than 1.

For the seven QTL associated with resistance to the CYR34 race at the seedling stage, 26 accessions possessed only one FH, while 39 accessions carried three to six FHs. In total, 49 accessions, all of which carried FHs, exhibited immunity to near immunity against the CYR34 race. To assess the relationship between the number of FHs and the disease response, we plotted the DS and IT values against the number of FHs (Appendix A). The analysis revealed a highly significant inverse correlation (*p* < 0.001), indicating that as the number of FHs increased, both the DS and the IT values decreased. These findings strongly suggest a cumulative effect of pyramiding favorable alleles, which significantly enhance the resistance of wheat to stripe rust.

Zhou8425B has been used as a founder parent in wheat breeding in China since its release in 1988 and exhibits durable adult-plant resistance to yellow rust [42]. In the present study, 18 accessions are the derivative varieties of Zhou8425B. With the exception of Zhengmai 158, the remaining 17 accessions carried at least one FH associated with resistance to the CYR34 race. Among them, Zhengpinmai 8, Guomai 9, and Fengdecunmai 1 also possessed FHs conferring resistance to the CYR33 race. Additionally, Zhengmai 1860 has an extra APR favorable haplotype related to adult-plant resistance (Appendix A).

## 3. Discussion

Due to the rapid breakdown of race-specific resistance, there is a growing consensus on the importance of breeding for durable resistance in crops. Pyramiding different resistance genes is an efficient strategy to develop varieties with durable resistance. In the past, this was nearly unfeasible because of the inability to detect different genes. However, molecular markers tightly linked to the target genes have now become powerful tools for this purpose. In the current study, we identified 14 QTL associated with stripe rust resistance, and their corresponding SNP markers will be invaluable for pyramiding resistance genes and breeding varieties with durable resistance. Additionally, we found that some varieties already carry multiple favorable alleles. For example, Zhengpinmai 8 carries seven FHs; these varieties may exhibit more durable resistance and could be selected as excellent resistant parents.

The SNP markers used for genotyping in this study are based on the 55K SNP array, which has been extensively used for the detection of agronomically important genes/QTL, with hundreds of studies published. Utilizing the same SNP markers facilitates the integration of the results, enabling the generation of a high-density map for agronomic traits and the identification of linked loci. For instance, the *QYr.baafs-6D* region was associated not only with yellow rust resistance but also with yield-related traits [43,44].

In this study, the *R*^2^ values of most QTL explaining phenotypic variances were less than 10%, lower than those detected in bi-parental populations. This phenomenon might be due to the relatively greater number of QTL identified here (a total of 14 QTL). Generally, the more QTL detected, the lower the *R*^2^ values tend to be. This is consistent with previous studies, where the percentage of the explained phenotypic variance (*R*^2^) of the associated markers was reported to range from 5 to 6% [45] and from 5 to 11% [46]. We observed some varieties carrying QTL with FHs but showing susceptibility to stripe rust, which might be due to background effects. Conversely, some accessions without FHs showed high resistance. For example, although Mazhamai showed immunity to CYR34, it did not possess any of the identified FHs, suggesting the presence of undetected resistance loci. To detect these missing QTL, the development of bi-parental populations is necessary. In addition, epistatic networks may suppress or enhance the effects of identified FHs. In some cases, recessive alleles at modifying loci could dampen the resistance expression in FH-carrying accessions, while synergistic interactions between minor-effect QTL might drive resistance in lines lacking major FHs.

It is noteworthy that we identified several accessions with FHs conferring resistance against stripe rust. Among them, Zhengmai 1860 showed resistance to both mixed *Pst* races and the CYR34 race. Additionally, seven accessions, including Guomai 9, Zhengpinmai 8, Shi 4185, Fengdecunmai 1, Yumai 13, Aikang 58, and Chuanmai 32, were resistant to both the CYR33 and CYR34 races. However, no accession was found to carry FHs against the CYR33, CYR34, and mixed *Pst* races simultaneously. Aikang 58 harbored the FHs of *QYr.baafs-2B.1* and *QYr.baafs-1D.1*, which conferred moderate resistance to the CYR33 race and immunity to the CYR34 race. These accessions can serve as valuable parental lines for future wheat breeding programs to enhance stripe rust resistance.

## 4. Materials and Methods

### 4.1. Plant Materials

A panel of 198 wheat accessions conserved in the Beijing Academy of Agricultural and Forest Sciences was used in this study. The panel comprised 173 accessions from China, 5 from Italy, 7 from the USA, 12 from Pakistan, and one from CIMMYT (Appendix A). The 173 Chinese wheat cultivars have been (or were) widely grown in the main wheat production zones since the early 20th century. The wheat cultivar Mingxian 169 served as a susceptible control.

### 4.2. Evaluation for Stripe Rust Resistance at the Seedling Stage

The 198 wheat cultivars were inoculated with the prevalent Chinese *Pst* races CYR33 and CYR34 at the seedling stage under controlled greenhouse conditions at the Plant Protection Institute of the Sichuan Academy of Agricultural Sciences, China. Five to six seeds of each cultivar were sown in a plastic pot filled with nutrient soil, with two replicates.

At the two-leaf stage (15 days after sowing), the seedlings were inoculated with urediniospores mixed with talc at an approximate ratio of 1:20. The inoculated plants were placed in a dew chamber at 10 ± 2 °C in darkness for 24 h and then transferred to a greenhouse at 18 ± 1 °C, with light supplementation of 10,000 lux and a photoperiod of 16 h light and 8 h darkness. The stripe rust reactions were recorded when the susceptible control, Mingxian 169, was fully infected, approximately 18 d post inoculation (dpi). The infection types (ITs) were scored using the 0–4 scale (0, 0;, 1, 2, 3, and 4), with the resistant (0–2) and susceptible (3–4) classifications as described previously [8]. The disease severity (DS) was scored as the percentage of the infected leaf area (0, 5, 10, 20, 40, 60, 80, or 100%) according to the rules for monitoring and forecasting wheat stripe rust according to GB/T 15795–2011 [47]. Accessions with a DS ≤ 20% were classified as resistance genotypes. Accessions with ITs of 0 to 0 and a DS < 1% were defined as immune or near immune according to GB/T 1443.1-2007 [25].

### 4.3. Evaluation of Stripe Rust Resistance at the Adult-Plant Stage

The APR of the 198 cultivars was evaluated in the field trials after artificial inoculation in three consecutive crop seasons (2018 to 2020) in Chengdu, Sichuan province (30°33′ N, 103°39′ E), hereafter abbreviated as CD18, CD19, and CD20, respectively. Each cultivar was sown in a 2.0 m row with 0.3 m between rows, and the sowing density was 30 seeds per row. The susceptible control, Mingxian 169, was planted every 20 rows, and spreader rows were planted to enhance the uniformity of the stripe rust inoculum across the trials. The field management followed local practices. At the tillering stage, the plants were artificially inoculated with a mixture of *Pst* races prevalent in China, including CYR32, CYR33, CYR34, Sull-4, Sull-5, Sull-7, and G22-14, as described previously [2]. The stripe rust scoring commenced when the disease severity on the flag leaves of Mingxian 169 reached 80%. The stripe rust symptoms were recorded three times at 7–10 day intervals. For each accession, the most severe disease response among the three scores was used for the analysis. The IT was estimated and classified, as at the seedling stage. Plants with an IT ≤ 2 and a DS ≤ 20% were classified as resistance genotypes, as described previously [8].

### 4.4. Genotyping and Genetic Analysis

The genomic DNA of the 198 cultivars was extracted from the leaves of two-week-old seedlings using a plant DNA kit (Biofit Co., Ltd., Beijing, China). A total of 53,063 probes from the wheat 55K SNP array (Affymetrix Axiom Wheat55K, Thermo Fisher Scientific Inc., Beijing, China) were used for the genotyping. Markers with missing values less than 10% and minor allele frequency (MAF) values greater than 5% were selected for subsequent analysis. The chromosomal positions of these SNPs were referred to the Chinese Spring reference genome (CS v1.0, http://202.194.139.32/, accessed on 1 January 2021).

The population structure was characterized using the Bayesian clustering algorithm implemented in STRUCTURE v2.3.4 [48] based on 44,759 polymorphic SNPs distributed on 21 wheat chromosomes. The number of presumed subpopulations (K) ranged from 2 to 10, with an admixture model and correlated allelic frequencies assumed. Ten independent STRUCTURE runs were performed with 50,000 replications for a burn-in and 10,000 replications for the Markov chain Monte Carlo (MCMC) analysis. The number of subpopulations and the best output were determined following the delta K method [49]. A neighbor-joining tree was constructed using software Tassel v3.0 and MEGA7 and visualized using the iTOL website (https://itol.embl.de/, accessed on 1 March 2021).

The genome-wide linkage disequilibrium (LD) was estimated as the squared allele frequency correlation (*r*^2^) between pairs of 44,759 SNP markers with known physical positions using TASSEL 3.0 [50]. The extent of the LD between pairs of loci was depicted with the *r*^2^ values against the inter-marker physical distance (Mb). Locally weighted polynomial regression curves were then fitted into the scatter plot. A marker–trait association analysis (MTA) was determined based on a mixed linear model (MLM) with the structure (Q) matrix as the fixed factor and the kinship (K) matrix as the random factor (Q+K, MLM) implemented in TASSEL 3.0 (https://www.thefreedictionary.com/tassel, accessed on 1 March 2021). The significant MTAs were selected using the criteria *p* ≤ 0.001 for all chromosomes at both the seedling and adult-plant stages. The high-confidence associated loci (*p* ≤ 0.01) flanking the significant MTA markers within the LD ≥ 0.3 decay distance region (about 5 Mb) on the same chromosome were considered as the QTL intervals. The MTAs detected in at least two environments at the adult-plant stage and in the two replications in the seedling trials, or detected at both the DS and IT trials, were considered as stable and reported in this study. Manhattan plots were generated using the CMplot package in the R program (https://github.com/YinLiLin/CMplot, accessed on 1 March 2021).

The QTL in this study were compared with previously reported Yr genes and QTL based on the physical positions of the significant markers using the Chinese Spring reference genome (CS v1.0, http://202.194.139.32/, accessed on 1 October 2021). Novel QTL were determined if they were not overlapping with previous Yr or QTL within a ±5.0 Mb interval.

## 5. Conclusions

This study identified seven elite wheat cultivars (five Chinese, one American, one Pakistani) exhibiting stable, high-level resistance (IT ≤ 2, DS ≤ 20%) against prevalent *Pst* races across the seedling and adult-plant stages. A GWAS revealed 14 robust QTL conferring resistance (11 ASR, 3 APR), including nine potentially novel loci. Within these QTL regions, favorable resistance haplotypes were defined, and key candidate genes (such as *TaPAP8* and *TaCRK2*) implicated in defense responses were identified. The integration of these validated resistant accessions, novel QTL, haplotype-specific markers, and functional gene candidates provides powerful molecular tools for efficiently pyramiding resistance genes, accelerating the development of durable wheat cultivars against stripe rust.

## Figures and Tables

**Figure 1 plants-14-01883-f001:**
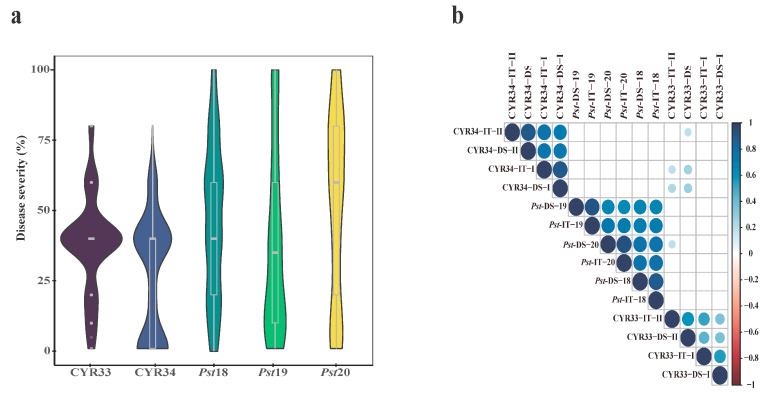
The distribution and correlation of the stripe rust responses at the seedling and adult-plant stages. (**a**) Violin plots illustrating the distribution of stripe rust resistance phenotypes for the CYR33 and CYR34 races at the seedling stage, and mixed *Pst* races at the adult-plant stage across three field environments (*Pst*18/19/20 represent the DS values scored at the CD18, CD19, and CD20 environments, respectively). (**b**) Heatmap of Pearson correlation coefficients quantifying the relationships among the stripe rust responses across the different races and developmental stages. The color intensity reflects the strength of the correlation, with positive (blue) and negative (red) values indicating direct or inverse associations, respectively. The phenotypic data consist of infection type (IT) and/or disease severity (DS) values, as defined in Section 4. “CYR33/34–DS/IT–I/II” represents the DS or IT values obtained after the inoculation with CYR33 or CYR34, which were scored during the first or second replicate in the greenhouse at the seedling stage.

**Figure 2 plants-14-01883-f002:**
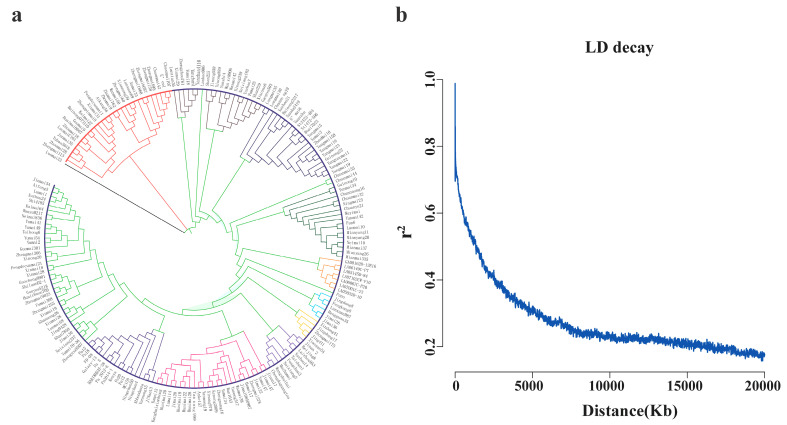
The phylogenetic relationships and linkage disequilibrium (LD) decay in 198 wheat accessions. (**a**) The neighbor-joining phylogenetic tree, constructed from the genome-wide single-nucleotide polymorphism (SNP) data, depicting the genetic relationships among the accessions. The branch colors correspond to the predefined groups (Appendix A). (**b**) A linkage disequilibrium decay plot based on 44,759 high-quality SNP markers. The LD is measured as the squared correlation coefficient (*r*²) between pairs of SNPs, plotted against the genomic distance (kilobase, Kb).

**Figure 3 plants-14-01883-f003:**
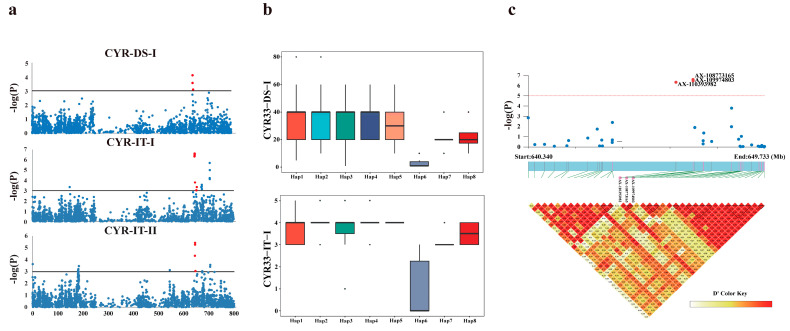
The effects of the *QYr.baafs-2B.1* haplotypes and candidate genes on the stripe rust resistance to the CYR33 race. (**a**) A Manhattan plot from a GWAS analyzing the seedling-stage responses to the CYR33 race. The genome-wide significance threshold is set at a –log10 (*p*) value of 3.0, with each dot representing an SNP marker. The red dots represent SNP markers significantly associated with *Pst* resistance. (**b**) The phenotypic effects of the *QYr.baafs-2B.1* haplotypes in the seedling-stage trials with the CYR33 race. The disease severity (DS) and infection type (IT) are shown as means ± standard deviation (SD). (**c**) A local Manhattan plot and haplotype block around *QYr.baafs-2B.1*. The zoomed region highlights the peak-associated SNPs (–log10 (*p*) ≥ 4.0) and their LD structure.

**Table 1 plants-14-01883-t001:** Stripe rust resistance QTL identified at both seedling and adult-plant stages.

QTL	Trait	Stage	Chr.	Position	Peak Marker	−log(*P*)	Marker *R*^2^	Region	FH	FH Frequency	QTL/Gene	Reference
				(Mb)			(%)	(Mb)				
*Yr.baafs-1B.1*	CYR34DS-I/II, CYR34IT-I/II	Seedling	1BS	56.89	AX-111170724	3.52–6.42	8.23–16.67	46.03–57.38	*Hap2*	0.202		
*Yr.baafs-1B.2*	CYR34DS-I/II, CYR34IT-I/II	Seedling	1BS	95.67	AX-94432535	3.23–6.93	8.84–18.32	90.55–96.88	*Hap2*	0.209		
*Yr.baafs-1B.3*	CYR34DS-I/II, CYR34IT-I/II	Seedling	1BS	229.31	AX-110084126	3.68–6.08	10.41–15.80	224.21–230.95	*Hap1*	0.213		
*Yr.baafs-1B.4*	20DS,19/20IT	Adult-plant	1BL	488.6	AX-108791049	3.00–3.72	7.44–9.25	487.94–491.08				
*Yr.baafs-1D.1*	CYR34DS-I/II, CYR34IT-I/II	Seedling	1DS	73.73	AX-94802245	3.04–6.24	7.68–17.54	59.53–84.62	*Hap3*	0.358		
*Yr.baafs-1D.2*	CYR34DS-I/II, CYR34IT-I/II	Seedling	1DS	109.82	AX-94747357	3.46–6.54	8.56–17.08	102.78–109.82	*Hap2*	0.189		
*Yr.baafs-2B.1*	CYR33IT-I/II, CYR33DS-I	Seedling	2BL	646.9	AX-108773165	3.05–6.60	7.61–17.16	643.69–655.52	*Hap6*	0.032	*QYrSM155.1*	[26]
*Yr.baafs-2B.2*	CYR33IT-I/II	Seedling	2BL	675.36	AX-111026674	3.01~4.05	7.54–10.33	673.05–675.79	*Hap7*	0.016		
*Yr.baafs-2B.3*	CYR33IT-I/II	Seedling	2BL	706.85	AX-111077615	3.26–5.70	8.65–14.82	706.72–707.18	*Hap4*	0.063		
*Yr.baafs-2B.4*	18DS,18IT	Adult-plant	2BL	785.83	AX-108900982	3.06–3.15	6.04–7.79	780.02–787.23			*qNV.Yr-2B.3*	[6]
*Yr.baafs-6B*	18/20DS,20IT	Adult-plant	6BS	90.4	AX-109912248	3.08–3.61	6.50–9.27	73.54–91.01	*Hap5*	0.042	*QYr.sicau-6BS*	[27]
											*QYr.nwafu-6BS.3*	[28]
*Yr.baafs-6D*	CYR34DS-I/II, CYR34IT-I/II	Seedling	6DL	467.52	AX-109362174	4.69–5.88	12.55–15.15	465.05–472.82	*Hap3*	0.202	*QYrCL.sicau-6DL*	[29]
*Yr.baafs-7B*	CYR33IT-I, CYR33DS-I	Seedling	7BS	64.33	AX-111143711	3.00–3.50	6.06–8.70	64.32–66.51				
*Yr.baafs-7D*	CYR34DS-II, CYR34IT-I/II	Seedling	7DS	104.49	AX-111855949	3.01–3.60	7.49–9.03	100.65–104.74			*QYr.rcrrc-7D*	[30]

## Data Availability

The original contributions presented in this study are included in the article/Appendix A. Further inquiries can be directed to the corresponding author.

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
