# Peer review of "Genome-Wide Identification of Wheat Gene Resources Conferring Resistance to Stripe Rust"

_plants, 2025, doi:10.3390/plants14121883_

Round 1

Reviewer 1 Report

Comments and Suggestions for Authors

The manuscript is well structured and presents a valuable dataset for identifying stripe rust resistance loci in wheat using a GWAS approach. The combination of phenotypic assessment at both seedling and adult stages, coupled with the identification of favorable haplotypes and candidate genes, adds significant depth to the study. The aims of this study are clear and the results are interest to me. I don’t have major comment. However, some clarifications and additions would improve the scientific rigour and utility of the findings.

First, while the authors describe certain accessions as “immune or nearly immune” to stripe rust, the criteria used to define such classifications (e.g., IT = 0, DS < 1%) are not explicitly stated in the methods. Given the reliance on DS and IT values for resistance evaluation, the threshold for categorizing accessions as immune should be clearly defined, with a reference to relevant standards (e.g., GB/T 15795–2011) or previous studies. This would help standardize the interpretation of results and enhance reproducibility.

Second, the authors note that several accessions carry favorable haplotypes (FHs) but still exhibit susceptibility, while others lacking identified FHs show high resistance. This observation is important and deserves further elaboration. While the text mentions the possibility of background genetic effects or undetected QTL, this point is somewhat underdeveloped. A brief discussion on the potential roles of epistasis, regulatory elements, or structural variants in influencing phenotypic expression would be beneficial.

Lastly, while the manuscript identifies multiple QTL as novel based on positional comparison, a clearer explanation of how novelty was assessed, especially in relation to overlapping confidence intervals and previously reported QTL, would strengthen the claim. If possible, including a summary table or supplementary figure cross-referencing the identified loci with prior studies would help contextualize these findings.

The English writing in the manuscript is overall understandable and technically sound. However, a number of grammatical and stylistic issues need to be addressed to improve clarity, fluency, and conciseness. Some sentences are overly long or repetitive, while others contain minor grammatical errors that affect readability. Below are detailed suggestions for revision:

Lines 43–44: please revise the phrase “due to its specific nature and the strong selection pressure exerted by,” to complete the clause. Suggested revision: "due to its specific nature and the strong selection pressure exerted by evolving pathogen populations, ASR is often rapidly overcome."

Lines 88–89: please revise the sentence “Nineteen accessions showed resistance to both Pst races, including 17 Chinese modern cultivars, one from USA (GA081628-13E16), and one from Pakistan (Pa 12).” to improve fluency. Suggested revision: "Nineteen accessions, including 17 modern Chinese cultivars, one from the USA (GA081628-13E16), and one from Pakistan (Pa12), exhibited resistance to both Pst races."

Lines 125–128: consider shortening and rephrasing for clarity. Suggested revision: "The 198 wheat accessions were grouped into two major clusters (G1 and G2) based on SNP data, with G1 further divided into two subgroups and G2 into eleven."

Lines 172–174: please revise the sentence “Accessions carrying haplotypes QYr.baafs-2B.1-Hap1 to QYr.baafs-2B.1-Hap5 accounted for 85.9% of the samples, with an average DS of 32.9% and an average IT of 2.8.” to improve clarity. Suggested revision: "Approximately 85.9% of the accessions carried haplotypes QYr.baafs-2B.1-Hap1 to Hap5, with mean DS and IT values of 32.9% and 2.8, respectively."

Lines 332–333: revise “Due to the rapid loss of race-specific resistance, there is a increasing consensus on the significance of breeding for durable resistance in crops.” for grammar and clarity. Suggested revision: "Due to the rapid breakdown of race-specific resistance, there is growing consensus on the importance of breeding for durable resistance in crops."

Lines 357–358: revise “Mazhamai exhibited immunity to CYR34 infection but lacked FHs from the CYR34 QTL, which could be due to undetected QTL.” to reduce ambiguity. Suggested revision: "Although Mazhamai showed immunity to CYR34, it did not possess any of the identified FHs, suggesting the presence of undetected resistance loci."

Lines 405–406: consider splitting the sentence “Stripe rust reactions were recorded three times, at 7 - 10 day intervals, and the most severe reaction to Pst races among the three recorded scores for each accession was used as the reaction of this accession in the analyses.” for better readability. Suggested revision: "Stripe rust symptoms were recorded three times at 7–10 day intervals. For each accession, the most severe disease response among the three scores was used for analysis."

Comments on the Quality of English Language

The English writing in the manuscript is overall understandable and technically sound. However, a number of grammatical and stylistic issues need to be addressed to improve clarity, fluency, and conciseness. Some sentences are overly long or repetitive, while others contain minor grammatical errors that affect readability. Below are detailed suggestions for revision:

Lines 43–44: please revise the phrase “due to its specific nature and the strong selection pressure exerted by,” to complete the clause. Suggested revision: "due to its specific nature and the strong selection pressure exerted by evolving pathogen populations, ASR is often rapidly overcome."

Lines 88–89: please revise the sentence “Nineteen accessions showed resistance to both Pst races, including 17 Chinese modern cultivars, one from USA (GA081628-13E16), and one from Pakistan (Pa 12).” to improve fluency. Suggested revision: "Nineteen accessions, including 17 modern Chinese cultivars, one from the USA (GA081628-13E16), and one from Pakistan (Pa12), exhibited resistance to both Pst races."

Lines 125–128: consider shortening and rephrasing for clarity. Suggested revision: "The 198 wheat accessions were grouped into two major clusters (G1 and G2) based on SNP data, with G1 further divided into two subgroups and G2 into eleven."

Lines 172–174: please revise the sentence “Accessions carrying haplotypes QYr.baafs-2B.1-Hap1 to QYr.baafs-2B.1-Hap5 accounted for 85.9% of the samples, with an average DS of 32.9% and an average IT of 2.8.” to improve clarity. Suggested revision: "Approximately 85.9% of the accessions carried haplotypes QYr.baafs-2B.1-Hap1 to Hap5, with mean DS and IT values of 32.9% and 2.8, respectively."

Lines 332–333: revise “Due to the rapid loss of race-specific resistance, there is a increasing consensus on the significance of breeding for durable resistance in crops.” for grammar and clarity. Suggested revision: "Due to the rapid breakdown of race-specific resistance, there is growing consensus on the importance of breeding for durable resistance in crops."

Lines 357–358: revise “Mazhamai exhibited immunity to CYR34 infection but lacked FHs from the CYR34 QTL, which could be due to undetected QTL.” to reduce ambiguity. Suggested revision: "Although Mazhamai showed immunity to CYR34, it did not possess any of the identified FHs, suggesting the presence of undetected resistance loci."

Lines 405–406: consider splitting the sentence “Stripe rust reactions were recorded three times, at 7 - 10 day intervals, and the most severe reaction to Pst races among the three recorded scores for each accession was used as the reaction of this accession in the analyses.” for better readability. Suggested revision: "Stripe rust symptoms were recorded three times at 7–10 day intervals. For each accession, the most severe disease response among the three scores was used for analysis."

Author Response

Comments 1: First, while the authors describe certain accessions as “immune or nearly immune” to stripe rust, the criteria used to define such classifications (e.g., IT = 0, DS < 1%) are not explicitly stated in the methods. Given the reliance on DS and IT values for resistance evaluation, the threshold for categorizing accessions as immune should be clearly defined, with a reference to relevant standards (e.g., GB/T 15795–2011) or previous studies. This would help standardize the interpretation of results and enhance reproducibility.

Response 1: Thank you for this important comment. We have clarified the criteria for defining "immune or near-immune" accessions in the Methods section (lines 402–403, 415–417) by explicitly referencing GB/T 1443.1–2007 (Agricultural Industry Standard of China). This standard defines: Immune as infection type (IT) = 0, Near-immune as IT = 0; (with a notation indicating trace infection). Additionally, we supplemented the disease severity (DS) threshold of < 1% based on phenotypic data, as accessions with IT = 0 or 0; consistently showed DS < 1% in our trials. This dual criterion (IT + DS) aligns with standard practices and enhances the reproducibility of resistance classifications.

Comments 2: Second, the authors note that several accessions carry favorable haplotypes (FHs) but still exhibit susceptibility, while others lacking identified FHs show high resistance. This observation is important and deserves further elaboration. While the text mentions the possibility of background genetic effects or undetected QTL, this point is somewhat underdeveloped. A brief discussion on the potential roles of epistasis, regulatory elements, or structural variants in influencing phenotypic expression would be beneficial.

Response 2: Thanks for your suggestion. We have added the potential roles of epistasis in discussion (Line 362-366).

Comments 3: while the manuscript identifies multiple QTL as novel based on positional comparison, a clearer explanation of how novelty was assessed, especially in relation to overlapping confidence intervals and previously reported QTL, would strengthen the claim. If possible, including a summary table or supplementary figure cross-referencing the identified loci with prior studies would help contextualize these findings.

Response 3: Thank you for this constructive suggestion. To strengthen the novelty assessment of identified QTL, we have implemented the following improvements: Novel QTL were strictly defined as those whose confidence intervals (±5.0 Mb) showed no physical overlap with previously reported Yr genes or QTL. This interval aligns with the genome-wide LD decay distance (r² ≥ 0.3) observed in our panel (Figure 2b), ensuring robust positional discrimination. The updated Methods (Lines 453-456) detail these criteria. We performed a comprehensive literature cross-referencing using the IWGSC RefSeq v1.0 genome coordinates: For formally designated Yr genes and SNP-based QTL, direct physical position comparisons were conducted. For studies reporting SSR-based QTL, we attempted to map markers by BLASTing SSR probe sequences against IWGSC v1.0. However, most of published SSR markers lacked probe sequences or yielded no unique hits to the reference genome. Given this limitation and to maximize reliability, comparisons prioritized SNP-anchored QTL from recent studies. All identified QTL overlapped with the QTL deteced in this study are now presented in Table 1.

Comments 4: The English writing in the manuscript is overall understandable and technically sound. However, a number of grammatical and stylistic issues need to be addressed to improve clarity, fluency, and conciseness. Some sentences are overly long or repetitive, while others contain minor grammatical errors that affect readability. Below are detailed suggestions for revision:

Lines 43–44: please revise the phrase “due to its specific nature and the strong selection pressure exerted by,” to complete the clause. Suggested revision: "due to its specific nature and the strong selection pressure exerted by evolving pathogen populations, ASR is often rapidly overcome."

Lines 88–89: please revise the sentence “Nineteen accessions showed resistance to both Pst races, including 17 Chinese modern cultivars, one from USA (GA081628-13E16), and one from Pakistan (Pa 12).” to improve fluency. Suggested revision: "Nineteen accessions, including 17 modern Chinese cultivars, one from the USA (GA081628-13E16), and one from Pakistan (Pa12), exhibited resistance to both Pst races."

Lines 125–128: consider shortening and rephrasing for clarity. Suggested revision: "The 198 wheat accessions were grouped into two major clusters (G1 and G2) based on SNP data, with G1 further divided into two subgroups and G2 into eleven."

Lines 172–174: please revise the sentence “Accessions carrying haplotypes QYr.baafs-2B.1-Hap1 to QYr.baafs-2B.1-Hap5 accounted for 85.9% of the samples, with an average DS of 32.9% and an average IT of 2.8.” to improve clarity. Suggested revision: "Approximately 85.9% of the accessions carried haplotypes QYr.baafs-2B.1-Hap1 to Hap5, with mean DS and IT values of 32.9% and 2.8, respectively."

Lines 332–333: revise “Due to the rapid loss of race-specific resistance, there is a increasing consensus on the significance of breeding for durable resistance in crops.” for grammar and clarity. Suggested revision: "Due to the rapid breakdown of race-specific resistance, there is growing consensus on the importance of breeding for durable resistance in crops."

Lines 357–358: revise “Mazhamai exhibited immunity to CYR34 infection but lacked FHs from the CYR34 QTL, which could be due to undetected QTL.” to reduce ambiguity. Suggested revision: "Although Mazhamai showed immunity to CYR34, it did not possess any of the identified FHs, suggesting the presence of undetected resistance loci."

Lines 405–406: consider splitting the sentence “Stripe rust reactions were recorded three times, at 7 - 10 day intervals, and the most severe reaction to Pst races among the three recorded scores for each accession was used as the reaction of this accession in the analyses.” for better readability. Suggested revision: "Stripe rust symptoms were recorded three times at 7–10 day intervals. For each accession, the most severe disease response among the three scores was used for analysis."

Response 4: Thanks for your meticulous reviewing and for identifying the language-related improvements. All suggested modifications have been carefully implemented in the main text. 

Reviewer 2 Report

Comments and Suggestions for Authors

see attachment

Author Response

Comments 1: Abbreviations: Abbreviations have to be explained either in the text or in Abbreviations list; e.g., Results, line 82, abbreviations for "IT" – "infection type" and "DS" – "disease severity" have to be explained here when used for the first time.

Response 1: Thank you for this feedback. We have addressed the abbreviation issue by ensuring that infection type (IT) and disease severity (DS) are defined at their first occurrence in both the Abstract and the Results section (line 80).

Comments 2: Terminology: In Results and in Discussion, lines 299, 317, the term „near immunity“ is used. It has to be briefly explained what does this term means.

Response 2: Thank you for this important comment. The term "near immunity" is defined in the Chinese agricultural standards GB/T 1443.1–2007 and GB/T 15795–2011 as an infection type (IT) scored as "0;". This designation refers to small necrotic spots on leaves without uredinia formation, indicating minimal pathogen development. The criteria align with international standards for partial resistance, where "0;" distinguishes near-immune responses from complete immunity (IT=0) while maintaining consistency in resistance classification.

Comments 3:Formal comments on the text related to English language and style:

In Abstract, lines 16, 22: Correct the typing error in the stripe rust race “CYR34” (not “CRY34”).

Abstract, line 13: Add „a“ prior to the word „key“ in the statement: „Breeding resistant varieties is a key to disease control.“

Introduction, line 36: Add a comma between the words „and“ and „in some cases“.

Introduction, line 63: Add a comma between „1997“ and „respectively“.

Results, line 111: Modify the word form „resistant“ to „resistance“ in the statement: „In total, 41 accessions exhibited resistance to the mixed Pst races (Table S1).“

Results, line 130: Add a space between „G1-1“ and „were“.

Results, line 166. Add a space between the words „QTL“ and „significantly“.

Results, line 176: Remove „the“ prior to „this study“ and remove „race“ prior to „CYR33 race“ in the statement: „….that most of the Chinese commercial wheat varieties used in this study were moderately susceptible to CYR33 race.“

Results, line 284: Add „a“ prior to the words „previous study“.

Materials and methods, line 397: Add a comma between the words „CD20“ and „respectively“.

Response3:Thank you for your meticulous review and for identifying the language-related improvements needed in the manuscript. All suggested modifications to the main text have been carefully implemented.

Reviewer 3 Report

Comments and Suggestions for Authors

Dear Authors, i have reviewed your manuscript.

you conducted a comprehensive genome-wide association study (GWAS) on 198 wheat lines to identify quantitative trait loci (QTL) and haplotypes conferring resistance to stripe rust (Puccinia striiformis f. sp. tritici) using a 55K SNP array. This work is novel in that it simultaneously assesses both seedling and adult-plant resistance (against CYR33, CYR34, and mixed Pst races), defines LD-based QTL blocks within fixed chromosomal regions, and pinpoints candidate genes. Their findings significantly advance wheat breeding by reporting 14 QTL—nine of which appear to be new loci—and by characterizing favorable allele combinations for durable resistance. 

Abstract: The abstract clearly states the study’s aims and principal findings, but it omits the statistical thresholds used and the exact sample sizes. It also fails to specify how many QTL are novel.

Introduction: The literature review is thorough and effectively contextualizes stripe rust biology and GWAS applicability. Nonetheless, certain historical disease details (e.g., specific Chinese epidemics) are overly detailed, while the manuscript’s central hypothesis is not explicitly articulated.

Conclusions:The manuscript lacks a standalone conclusion. I recommend adding a concise, bullet-pointed Conclusion that highlights the key discoveries, their practical implications, and any methodological recommendations.

Author Response

Comments 1: The abstract clearly states the study’s aims and principal findings, but it omits the statistical thresholds used and the exact sample sizes. It also fails to specify how many QTL are novel.

Response 1: Thank you for this critical feedback. We have made revision in the abstract to include the following details: 1) as for the sample size and statistical thresholds: “198 modern wheat varieties were phenotyped with prevalent Pst races CYR33 and CYR34 at the seedling stage and with mixed Pst races at the adult-plant stage. Seven stable resistance varieties with infection type (IT) ≤ 2 and disease severity (DS) ≤ 20%......Genome-wide association study (GWAS) identified 14 QTL using a significance threshold of P ≤ 0.001”; 2) as for the novel QTL, “Nine of these QTL were potentially novel, as they did not overlap with previously reported Yr or QTL loci within a ±5.0 Mb interval (consistent with genome-wide LD decay)” in abstract. We also added Novel QTL clarification in Methods (Lines 453-456). 

Comments 2: Introduction: The literature review is thorough and effectively contextualizes stripe rust biology and GWAS applicability. Nonetheless, certain historical disease details (e.g., specific Chinese epidemics) are overly detailed, while the manuscript’s central hypothesis is not explicitly articulated.

Response 2: Thank you for this insightful feedback. We have revised the Introduction to address your concerns for the overly detailed historical disease by condensing to a concise statement: "Newly evolved Pst races (e.g., CYR34) exhibit broader virulence spectra, overcoming most known resistance genes [14-16]."

Comments 3: Conclusions: The manuscript lacks a standalone conclusion. I recommend adding a concise, bullet-pointed Conclusion that highlights the key discoveries, their practical implications, and any methodological recommendations.

Response 3: Thank you for this important suggestion. We have added a standalone Conclusion section structured with bullet points to summarize the study’s impact.

Round 2

Reviewer 3 Report

Comments and Suggestions for Authors

I recommand it for publication.